# An In-Context Learning Theoretic Analysis of Chain-of-Thought

**Chenxiao Yang**[1]  **Zhiyuan Li**[1]  **David Wipf**[2]

## Abstract

Large language models (LLMs) have demonstrated remarkable reasoning capabilities with proper prompting strategies such as by augmenting demonstrations with chain-of-thought (CoT). However, the understanding of how different intermediate steps in the CoT improve reasoning and the principles guiding their design remains elusive. This paper takes an initial step towards addressing these questions by introducing a new analytical framework from a learning theoretic perspective. Particularly, we identify a class of in-context learning (ICL) algorithms on few-shot CoT prompts, capable of learning complex nonlinear functions by composing simpler predictors obtained through gradient descent based optimization. We show this algorithm can be expressed by Transformers in their forward pass with simple weight constructions. We further analyse of the generalization properties of the ICL algorithm for learning different families of target functions. The derived theoretical results suggest several provably effective ways for decomposing target problems and forming CoT prompts, highlighting the bottleneck lies at the hardest reasoning step. Empirically, we demonstrate that CoT forms derived from our theoretical insights significantly enhance the reasoning capabilities of real-world LLMs in solving challenging arithmetic reasoning tasks.

## 1. Introduction

Large language models (LLMs) have demonstrated remarkable success in a variety of reasoning tasks, such as arithmetic, commonsense and logical reasoning (Cobbe et al., 2021; Rae et al., 2021; Srivastava et al., 2022). An abundance of empirical evidences ((Brown et al., 2020; Wei et al., 2022; Kojima et al., 2022; Liu et al., 2021; Zhou

et al., 2022; Wang et al., 2023; Yao et al., 2024; Besta et al., 2024; Madaan et al., 2024), inter alia) suggest that their effectiveness to handle these complex tasks hinges significantly on the design of prompts. Particularly, given few-shot examples in the form of question-answer pairs, LLMs can learn new tasks in-context – tasks that were not explicitly seen during their training (Brown et al., 2020; Garg et al., 2022). Since its introduction, the so-called in-context learning (ICL) phenomenon has intrigued a long line of work attempting to mechanistically explain it ((Akyürek et al., 2023; Chan et al., 2022; Olsson et al., 2022; Xie et al., 2021; Von Oswald et al., 2023), inter alia), or proposing new methods for enhancement.

One notable method is called chain-of-thought (CoT) (Wei et al., 2022; Reynolds & McDonell, 2021; Nye et al., 2021). Few-shot CoT augments in-context examples with intermediate steps, which have been shown highly effective for enhancing the performance of LLMs on a wide variety of reasoning tasks. Several theoretical works have explored the benefits of CoT from the perspectives of expressiveness (Li et al., 2024; Feng et al., 2024), demonstrating families of circuits that can or can not be expressed by Transformers with and without CoT. Additionally, its effectiveness has also been case studied by in-context learning ReLU MLPs, where intermediate steps in the prompts are internal representations of MLP (Li et al., 2023b). Despite these advancements, the theoretical understanding of CoT is far from complete, particularly in terms of how specific reasoning steps function for certain tasks and the principles for designing effective CoT prompts. Therefore, this paper takes an initial step towards answering the open question:

*How do intermediate steps in CoT prompting affect the ability of LLMs to learn complex reasoning tasks?*

This question is important not only because of its potential to uncover the inner workings of CoT but also because it could bridge theory and practice by informing effective prompting design. In fact, designing better prompts is a central aspect of prompt engineering, and most existing methods are from heuristics (Fu et al., 2023b; Shao et al., 2023; Prasad et al., 2023). The main challenge of theoretically answering this question lies in the limited knowledge about the computations that pre-trained Transformers actually perform, and the lack of ways to confirm hypotheses

[1]Toyota Technological Institute at Chicago [2]Amazon Web Services. Correspondence to: Chenxiao Yang <chenxiao@ttic.edu>.

*Proceedings of the 1st Workshop on In-Context Learning at the 41st International Conference on Machine Learning*, Vienna, Austria. 2024. Copyright 2024 by the author(s).

(if any) due to their black-box nature. Since without this knowledge, analysing the behavior of models on different prompts would be largely intractable, we propose a learning theoretic framework, which first uncovers a class of ICL algorithms Transformers can express with CoT, and then derive results in regime and maps them onto reality. Therefore, this paper's contributions are:

- **Section 3: Identify a class of ICL algorithms enabled by Transformers with CoT**. This class of algorithms take as input CoT examples, and output a compositional non-linear predictor $h$ by composing predictors $\{h_i\}_{i \in [k]}$ obtained from $k$ sub-algorithms. Each sub-algorithm attends to a reasoning step in the CoT, and extracts the corresponding current/next reasoning step pairs to learn a linear predictor on top of a non-linear feature map, i.e. $h_i : x \mapsto W_i \phi_i(x)$. By recurrently stacking non-linearity and linearity, these predictors collectively form a highly complex non-linear function. To validate the consideration of these algorithms, Theorem 3.1 establishes that, Transformers can express this algorithm class in an end-to-end manner, producing a non-linear predictor $h$ in a single forward pass. We do so by showing that the gradient descent dynamics of all intermediates results and the output can be simultaneously simulated by self-attention with simple parameter configurations.

- **Section 4: Analyse generalization of ICL algorithms w.r.t. different forms of CoT**. Given a reasoning task involving a family of target functions, we formalize the operation of "decomposing the task" and identify specific properties of the decomposition that are effective (or ineffective) for enhancing the ICL algorithm's generalization performance. Theorem 4.2 provides an upper bound for the expected error of the ICL algorithm with CoT, indicating that the difficulty of learning is determined by the hardest (i.e. least sample-efficient) CoT step after task decomposition. Practically, this means that a provably effective way to design a chain is by reducing the hardness of the hardest step. Furthermore, Theorem 4.3 establishes a lower bound for the ICL algorithm without CoT or with suboptimal CoT. Applying these bounds to specific tasks leads to concrete recommendations for designing CoT. For example, by case studying learning parities, we show that introducing a specific intermediate reasoning step guarantees smaller errors for the ICL algorithm, given sufficient samples.

- **Section 5: Corroborate theoretical predictions on real-world LLMs**. In particular, we propose a methodology to construct complex reasoning tasks with varying overall hardness. This is achieved by incrementally composing random elementary functions (e.g. basic arithmetic operations $+, -, \times$) to create an increasingly difficult target function (e.g. polynomials). We reveal only a limited number of intermediate results in the CoT prompting to control the difficulty of the hardest reasoning step. We observe that the

success rates of LLMs improve significantly as the difficulty of the hardest step in the CoT reduces, regardless of the overall task complexity. Additionally, on two notoriously difficult-to-learn Boolean function tasks, parities and DNFs, our empirical results demonstrate that CoT forms derived from our analyses significantly enhance reasoning performance, sometimes improving accuracy from nearly random guessing to nearly perfect. These results demonstrate a close alignment between our learning theoretic predictions and practical prompting designs.

Finally, in **Section 7**, we remark that the open research question studied in this paper is inherently challenging, and there exist several limitations regarding the simplifications made for analytical tractability. We also point out several directions to extend the analysis and broaden its use cases.

## 2. Preliminary

**In-Context Learning (ICL).** In ICL (Garg et al., 2022), the base model is provided with $N$ demonstrations (a.k.a. in-context examples) $e^{(i)} = (x^{(i)}, y^{(i)})$ for $i \in [N]$ where $x \in \mathcal{X}$ is the input and $y \in \mathcal{Y}$ is the output. The goal of the model is to learn an unknown target function $f \in \mathcal{F} : \mathcal{X} \to \mathcal{Y}$. The learning process is conducted by an algorithm denoted as $A : (\mathcal{X} \times \mathcal{Y})^N \to \mathcal{H}$, which takes demonstrations as input and outputs a predictor (a.k.a. hypothesis) $h : \mathcal{X} \to \mathcal{Y}$ from a hypothesis class $\mathcal{H}$ defined by the learning algorithm $A$. The model then uses this predictor to make predictions on a query input $x^{(N+1)}$ (which we assume is from the same distribution as demonstrations). For instance, if the base model is a pre-trained Transformer, denoted as $\mathrm{TF}_\theta(\cdot)$, ICL can be described as

$$\begin{aligned}
&\mathrm{TF}_\theta(\{(x^{(i)}, y^{(i)}) : i \in [N]\}, x^{(N+1)}) \\
&= A(\{(x^{(i)}, y^{(i)}) : i \in [N]\}; \mathcal{H})(x^{(N+1)})
\end{aligned} \tag{1}$$

where $h = A\left(\{(x^{(i)}, y^{(i)}) : i \in [N]\}; \mathcal{H}\right)$ is the predictor learned by the Transformer in-context.

**Transformers.** Let $E = \{e^{(i)} : i \in [N]\} \in \mathbb{R}^{d \times N}$ denote the concatenation of demonstrations, and let $e^{(N+1)} = (x^{(N+1)}, 0) \in \mathbb{R}^d$ whose dimension aligns with other examples. Following the setup in previous works (Von Oswald et al., 2023; Ahn et al., 2023; Cheng et al., 2024; Zhang et al., 2023), a single-head self-attention layer with fixed weights $W_K, W_Q, W_V \in \theta$ updates $e^{(N+1)}$ as

$$e^{(N+1)} \leftarrow e^{(N+1)} + W_V E \sigma(E^\top W_K^\top W_Q e^{(N+1)}), \tag{2}$$

where $\sigma$ is non-linearity that could be specified as Softmax, ReLU or some kernel functions, e.g. (Choromanski et al., 2021; Katharopoulos et al., 2020; Wang et al., 2020; Peng et al., 2021). Stacking multiple self-attention layers (optionally with the MLP module) gives us the Transformer

considered in this paper. Note that we exclude the query token when computing the attention, following (Von Oswald et al., 2023). We consider this for better illustrating our construction and can be relaxed.

**Chain-of-Thought (CoT).** CoT (Wei et al., 2022) prompting instructs the model to solve a problem step-by-step. In this paper, we consider the setup in the seminal work, i.e. few-shot CoT, where both demonstration and prediction are in the form of CoT. In particular, for $k$ reasoning steps,[1] we denote each demonstration as $e = (x, z_1, \cdots, z_{k-1}, y)$, where $z_t \in \mathcal{Z}_t$ represents $t$-th intermediate reasoning step. Let $d$ be the dimension of $e$ and $d(\mathcal{Z})$ the dimension corresponding to a certain space $\mathcal{Z}$. For convenience, we also let $z_0 = x$ and $z_k = y$. While this paper focuses on few-shot CoT, our insights could potentially also generalize to other settings such as zero-shot CoT and CoT in pre-training.

## 3. In-Context Learning with CoT

### 3.1. In-Context Learning Algorithms

We begin by introducing a new class of ICL algorithms $A$ that is enabled by CoT prompting. The algorithm is composed of several sub-algorithms and learns increasingly complex compositional functions with more CoT steps. The algorithm is described as follows:

---
**Algorithm 1** ICL with CoT
---
**Input:** Examples $\{(x^{(j)}, z_1^{(j)}, \cdots, z_{k-1}^{(j)}, y^{(j)})\}_{j=1}^N$, Learning Algorithms $\{A_i\}_{i=1}^k$.
**Output:** Predictor $h : \mathcal{X} \to \mathcal{Y}$
**for** $i = 1, \cdots, k$ **do**
$\quad \lfloor \; h_i \leftarrow A_i(\{(z_{i-1}^{(j)}, z_i^{(j)})\}_{j=1}^N; \mathcal{H}_i)$
$h \leftarrow h_k \circ \cdots \circ h_2 \circ h_1$

---

This class of ICL algorithms take as input a set of demonstrations, where each demonstration contains $k$ reasoning steps, and outputs a predictor $h$ from $\mathcal{X}$ to $\mathcal{Y}$. The learning is performed in a step-by-step manner; that is, for each reasoning step $i \in [k]$, a sub-algorithm $A_i$ is used to learn a predictor $h_i \in \mathcal{H}_i : \mathcal{Z}_{i-1} \to \mathcal{Z}_i$ where $\mathcal{H}_i$ is a hypothesis space associated with $A_i$. The learned predictors $h_1, h_2, \cdots, h_k$ are then composed to obtain the desired overall predictor. Although this class of ICL algorithms stipulates that each step is derived from the previous step, it can be naturally extended to scenarios where each step is a function of all preceding steps. For instance, we can redefine $z_i$ in the algorithm as a concatenation of $\{z_j\}_{j \leq i}$ in the initial CoT. Therefore, without loss of generality, we assume that the

---
[1]We also treat the output as a reasoning step. Thus, there is at least one step even without CoT.

CoT satisfies the Markov property, meaning that each step is conditionally dependent only on the immediately preceding step.

While this new class of ICL algorithms could have many different variants, in this paper, we are interested in each $A_i$ defined as empirical risk minimization – more specifically – using gradient descent to minimize a loss $\mathcal{L}_i$ over in-context examples to learn a predictor from the hypothesis class $\mathcal{H}_i$. The loss function is defined as the squared loss, and the hypothesis class is defined as a linear function class on fixed features, i.e. $\mathcal{L}_i = \frac{1}{2} \sum_{j=1}^N \|h_i(z_{i-1}^{(j)}) - z_i^{(j)}\|_2^2$ and

$$h_i \in \mathcal{H}_i = \{z_{i-1} \mapsto W_i \phi_i(z_{i-1}) : \|W_i\|_2 \leq B\}, \quad (3)$$

where $\phi_i : \mathcal{Z}_{i-1} \to \mathbb{R}^K$ is a (possibly non-linear) feature map to a $K$-dimensional latent space, and $W_i \in \mathbb{R}^{d(\mathcal{Z}_i) \times K}$ are learnable weights whose norm is bounded by $B$. Intriguingly, while each hypothesis class corresponding to each $A_i$ is a linear function class on fixed features, the predictor obtained from the ICL algorithm is a non-linear function, which can be written as stacking multiple non-linearities and linear transformations, i.e.

$$h = W_k \phi_k(\cdots(W_2 \phi_2(W_1 \phi_1(x)))) \in \mathcal{H}. \quad (4)$$

For example, if $\phi_1$ is specified as the identity mapping, and $\phi_i$ for $i \neq 1$ are specified as $\mathrm{ReLU}$, $\mathrm{Tanh}$, or other activation functions, (4) could represent a $k$-layer deep neural network. Beyond this, the ICL algorithm is highly flexible in terms of the range of functions that can be in-context learned with CoT. As the number of intermediate steps increases, the predictor also becomes more powerful. Moreover, as will be detailed in the next subsection, such a powerful algorithm is particularly compelling for analysis since it can be expressed by a Transformer with simple weight constructions.

### 3.2. Transformers Learn Non-Linear Functions

Next, we demonstrate the construction of parameters that enable Transformers to express the ICL algorithm in their forward pass. Recently, some works show the inherent connection between the self-attention layer and the dynamics of gradient descent (GD) optimization (Akyürek et al., 2023; Von Oswald et al., 2023; Dai et al., 2023; Ahn et al., 2023; Mahankali et al., 2023; von Oswald et al., 2023; Zhang et al., 2023; Cheng et al., 2024). To illustrate this more clearly, consider a simplified case without CoT (i.e. $k = 1$). The loss is $\mathcal{L} = \sum_{i \in [N]} \|h(x^{(i)}) - y^{(i)}\|_2^2 / 2$, and GD with a fixed step size updates the weight as $W \leftarrow W - \eta \nabla_W \mathcal{L}$. This process also induces dynamics in function space, i.e. the evolution of the learned function $h$ as weights update. For linear function class in (3) and prediction on the query input, the function space dynamics are as follows (see deriva-

tion in Appendix A.1), $h(x^{(N+1)})$

$$\leftarrow \underbrace{h(x^{(N+1)})}_{\text{Last-Step Prediction}} + \eta \underbrace{(Y - \hat{Y})}_{\text{Residuals}} \underbrace{\phi(X)^\top \phi(x^{(N+1)})}_{\text{Kernel Function}} \quad (5)$$

where $Y = [y^{(i)}]_{i=1}^N \in \mathbb{R}^{d(\mathcal{Y}) \times N}$, $\hat{Y} = [h(x^{(i)})]_{i=1}^N$, $\phi(X) = [\phi(x^{(i)})]_{i=1}^N \in \mathbb{R}^{K \times N}$. The residuals $Y - \hat{Y}$ refer to the difference between labels and predictions, which is equivalent to $Y$ at initialization if the weights in $h$ are initialized as 0. The last term represents a kernel function w.r.t. the feature map $\phi$, quantifying the similarity between the test (i.e. query) and training examples (i.e. demonstrations).

For comparison, we also rewrite the self-attention layer, where we reinterpret $e^{(i)}$ in the forward pass as a concatenation of input and the residual $(x^{(i)}, y^{(i)} - h(x^{(i)}))$, which at initialization is equivalent to $(x^{(i)}, y^{(i)})$. In particular, we have $e^{(N+1)}$

$$\leftarrow \underbrace{e^{(N+1)}}_{\text{Skip Connection}} + \underbrace{W_V E}_{\text{Embedding}} \underbrace{\sigma(E^\top W_K^\top W_Q e^{(N+1)}))}_{\text{Attention Module}} \quad (6)$$

where the last term is the attention module, also quantifying certain similarities between the query and demonstrations. Let us show how (6) subsumes (5) under certain weight constructions and Transformer architectural choices. For linear self-attention layer without activation $\sigma$ in (6), the weight constructions $W_V = \begin{pmatrix} 0_{d(\mathcal{X})} & 0 \\ 0 & -\eta I_{d(\mathcal{Y})} \end{pmatrix}$ and $W_K^\top W_Q = \begin{pmatrix} I_{d(\mathcal{X})} & 0 \\ 0 & 0_{d(\mathcal{Y})} \end{pmatrix}$ yields

$$\begin{aligned} E^\top W_K^\top W_Q e^{(N+1)} &= X^\top x_{N+1}, \\ W_V E &= -\eta(0_{d(\mathcal{X})}, Y - \hat{Y}), \end{aligned} \quad (7)$$

which is equivalent to (5) if $\phi$ is the identity mapping; that is, Transformers can perform linear regression in their forward pass (Von Oswald et al., 2023). More generally, the connections between kernel function and attention module have been widely studied, e.g. (Tsai et al., 2019; Wright & Gonzalez, 2021; Chen et al., 2024), allowing us to apply similar reasoning to kernel regression to establish connections between (5) and (6). Such an extension has also been discussed in a concurrent work (Cheng et al., 2024), optionally using more sophisticated constructions (Guo et al., 2024), and empowered by the MLP modules (Von Oswald et al., 2023) (Proposition 2); we refer interested readers to these works for details. In practice, kernelized attentions (Choromanski et al., 2021; Katharopoulos et al., 2020; Wang et al., 2020; Peng et al., 2021) are very effective and have been widely used due to their superior linear complexity.

**Our Construction.** Next, we present our construction of Transformers that allows them to learn compositional functions in-context by implementing the ICL Algorithm 1 in their forward pass.

**Theorem 3.1.** *Given a set of in-context examples in the form of CoT with $k$ reasoning steps (as defined in Section 2) and a query input $x^{(N+1)}$, Transformers with depth $\mathcal{O}(kt)$ can express Algorithm 1 where $A_i$ is $t$ steps of GD on squared loss and $\mathcal{H}_i$ is a linear function class defined in (3) whose feature map aligns with the attention.*

We provide a high-level sketch of the construction here and defer details to Appendix A.1. First, we define $k$ loss functions $\{\mathcal{L}_i\}_{i \in [k]}$ associated with $k$ reasoning steps. Each loss function is convex w.r.t. the weights of the corresponding predictor. In the forward pass, similar with (5), Transformers implement (kernel) GD dynamics in function space to minimize these loss functions. One challenge of establishing the connection between (5) and (6) by further considering CoT is that, for compositional non-linear predictors $h$ in (4), updating weights in a prior-step predictor (e.g. $W_1$ in $h_1$) could cause non-linear dynamics in the final prediction $h(x)$ in (4). We show that this can be circumvented if the learning is done in a step-by-step manner, namely Transformers first learn a preceding reasoning step using their lower layers, then proceed to learn the next step using upper layers.

Our construction subsumes the constructions in Von Oswald et al. (2023) and Cheng et al. (2024) as special cases where $k = 1$. The construction is not unique and could be adapted to other setups, such as recurrently making $k$ predictions (Li et al., 2023b) in $k$ forward passes. Compared with (Li et al., 2023b), ours is more general, arguably simpler, and yields more powerful predictors beyond ReLU MLPs.

## 4. Learning Theoretic Principle of Prompting

### 4.1. Quantifying the Benefit of CoT

As we have shown, CoT enables Transformers to in-context learn compositional non-linear predictors. A natural follow-up question arises: will ICL succeed given different forms of CoT prompting, and how can we design intermediate results to improve the reasoning capabilities of LLMs? To answer these questions, we investigate the generalization properties of the ICL algorithm and link them to the task distributions from which the demonstrations are generated.

Formally, let $\mathcal{D}$ be a distribution over input space $\mathcal{X}$, and $f : \mathcal{X} \to \mathcal{Y}$ a fixed target function. An input distribution and target function pair $(f, \mathcal{D})$ defines the generating process of in-context examples, where examples are drawn based on $x \sim \mathcal{D}(x)$, $y = f(x)$. Let $\mathcal{P}$ be a family of distributions defined as a set of $(f, \mathcal{D})$ pairs, representing a certain reasoning task where the target function is typically not unique. The following error quantifies how successfully the ICL algorithm can learn the task

$$\Delta(\mathcal{P}, h) \triangleq \max_{(f, \mathcal{D}) \in \mathcal{P}} \mathbb{E}_{x \sim \mathcal{D}(x)} \left[ l\left(h(x), f(x)\right) \right] \quad (8)$$

where $l$ is the squared loss and could be extended to other convex loss functions, $h$ is the predictor obtained from the ICL algorithm on $N$ i.i.d. examples from $(f, \mathcal{D})$. Minimizing the error guarantees the performance of ICL across all possible target functions within the family.

**Effects of CoT.**  Given a fixed reasoning task $\mathcal{P}$, and an ICL algorithm $A$, the remaining crucial factor determining $\Delta(\mathcal{P}, h)$ is how we decompose the target function. Different decompositions yield different CoT examples, resulting in different predictors $h$ with potentially distinct performances. To formalize this, suppose there exists an operator $T$ such that for each $(f, \mathcal{D}) \in \mathcal{P}$, it decomposes the target function as $T(f) = (f_2, f_1)$, $f = f_2 \circ f_1$, where $f_1 : \mathcal{X} \to \mathcal{Z}$ and $f_2 : \mathcal{Z} \to \mathcal{Y}$ for another space $\mathcal{Z}$. This decomposition induces two new distribution families

$$\begin{aligned} \mathcal{P}_{T,1} &= \{\!\{(f_1, \mathcal{D}) : (f_2 \circ f_1, \mathcal{D}) \in \mathcal{P}\}\!\} \\ \mathcal{P}_{T,2} &= \{\!\{(f_2, \mathcal{D}') : (f_2 \circ f_1, \mathcal{D}) \in \mathcal{P}\}\!\} \end{aligned} \quad (9)$$

where $\{\!\{\cdot\}\!\}$ denotes multiset that allows possibly repeating elements, and $\mathcal{D}' : \mathcal{Z} \to \mathbb{R}$ is determined by $f_1$ and $\mathcal{D}$. The decomposition always exists and is not unique (e.g. $f_1$ could be any bijection). Particularly when $f_1$ is specified as the identity mapping, we have $\mathcal{P}_{T,2} = \mathcal{P}$. For each decomposition, we can associate it with the generating process of demonstrations, i.e. $x \sim \mathcal{D}(x)$, $z = f_1(x)$, $y = f_2(z)$.

Stipulating intermediate steps to be generated by certain $f_1$ implicitly assumes: 1) no additional input information is required to get the intermediate step, and 2) the intermediate step is sufficiently informative for predicting the label. Note that while we consider only one intermediate step here, all of the following results can be iteratively applied to accommodate CoT with more steps.

## 4.2. Theoretical Results

In this section, we analyse the ICL error $\Delta(\mathcal{P}, h)$ w.r.t. different task decomposition $T$, aiming to: 1) identify properties of the decomposition that are favorable for the ICL performance, and 2) elucidate how well-designed intermediate results in CoT can guarantee better performance on certain reasoning tasks. Proofs are deferred to Appendix A.

**Upper Bound.**  First, we present an upper bound for $\Delta(\mathcal{P}, h)$.

**Lemma 4.1.** *For any distribution family $\mathcal{P}$ and decomposition operator $T$, the ICL Algorithm 1 on CoT demonstrations sampled from the corresponding distributions in $\mathcal{P}_{T,1}$ and $\mathcal{P}_{T,2}$ has an error upper bound*

$$\Delta(\mathcal{P}, h) \leq 4c_{B,\phi} \max\left\{\Delta(\mathcal{P}_{T,1}, h_1), \Delta(\mathcal{P}_{T,2}, h_2)\right\} \quad (10)$$

*where $c_{B,\phi} = \max\{1, B^2 \operatorname{Lip}(\phi)^2\}$ is a constant determined by hypothesis class $\mathcal{H}$ in (3), $h_1$ and $h_2$ are predictors given by sub-algorithms $A_1$ and $A_2$ such that $h = h_2 \circ h_1$.*

Note that the result also applies to other sub-algorithms $A_1$ and $A_2$ beyond linear/kernel regression (as long as the loss function is convex). Iteratively applying Lemma 4.1 allows us to derive an upper bound for $\Delta(\mathcal{P}, h)$ with $k$-step CoT enabled by multiple iterations of decomposition.

**Theorem 4.2.** *For any distribution family $P$ and decomposition operators $\{T_i\}_{i \in [k-1]}$ sequentially applied on $\mathcal{P}$, the ICL Algorithm 1 with $k$-step CoT has an error upper bound*

$$\Delta(\mathcal{P}, h) = \mathcal{O}(\max_{i \in [k]} \{\Delta(\mathcal{P}_{T,i}, h_i)\}) \quad (11)$$

*where $\mathcal{P}_{T,i}$ is the induced distribution family that generates the $i$-th reasoning step, and $h_i$ is the predictor given by sub-algorithm $A_i$.*

This result shows that the largest error, i.e. $\max_i \{\Delta(\mathcal{P}_{T,i}, h_i)\}$, made by predictor $h_i$ at the hardest reasoning step, i.e. $\operatorname{argmax}_i \{\Delta(\mathcal{P}_{T,i}, h_i)\}$, is the bottleneck for the eventual performance of ICL. To minimize $\Delta(\mathcal{P}, h)$, it suffices to minimize the error made at the hardest step.  Particularly, suppose each step is PAC learnable,[2] the hardest reasoning step is exactly the least sample-efficient one (i.e. needing the highest rate of sample number to achieve an error $\epsilon$) for large enough $N$. Therefore, in CoT, it is desirable to decompose the hardest step, since by doing so, ICL can have better performance guarantee and correspondingly Transformer also has the expressiveness to achieve the desired error (based on results in Section 4). The practical lesson for designing CoT from Theorem 4.2 is: **an effective way to form a CoT is by decomposing the hardest or the most sample inefficient reasoning step into smaller steps that are easier to learn**.

Our results align with existing empirical practices of decomposing initially challenging tasks into sub-tasks, e.g. (Zhou et al., 2022; Khot et al., 2022; Zhang et al., 2022), and complement these works by offering a well-founded (and also experimentally supported) approach to quantify different task decompositions.

**Lower Bound.**  Moreover, we demonstrate negative cases where the ICL algorithm could fail to solve certain challenging reasoning tasks without CoT or with suboptimal CoT. We achieve this by establishing a lower bound for $\Delta(\mathcal{P}, h)$ under undesirable conditions that should be avoided.

**Theorem 4.3.** *For any distribution family $\mathcal{P}$ and decomposition operator $T$, suppose ICL Algorithm 1 returns a first-step predictor $h_1$ from a finite set $\mathcal{H}'_1 \subseteq \mathcal{H}_1$, the ICL error has lower bound*

$$\Delta(\mathcal{P}, h) \geq \frac{1}{2} - B\sqrt{K|\mathcal{H}'_1|\operatorname{Var}(\mathcal{P})} \quad (12)$$

---

[2]PAC learnability (Shalev-Shwartz & Ben-David, 2014) means there exists a learning algorithm that, with a sufficient number of examples, can return a predictor in $\mathcal{H}$ that achieves an arbitrarily small error $\epsilon$ with high probability $1 - \delta$.

*where $K$ is the number of features for the non-linearity $\phi$, and $\mathrm{Var}(\mathcal{P})$ depends on $\mathcal{P}$; we defer its definition to Appendix A.3.*

In the extreme case $|\mathcal{H}_1'| = 1$ or no CoT, this becomes the lower bound for linear/kernel regression (Malach & Shalev-Shwartz, 2022). This result suggests that, given a certain task, the benefit from decomposition $T$ would be compromised if it causes the ICL algorithm to output a limited set of $h_1$ (i.e. when $|\mathcal{H}_1'|$ is small). This helps rule out some dummy cases, such as when the first step of CoT is the identity mapping. The term $\mathrm{Var}(\mathcal{P})$ is associated with the distribution family $\mathcal{P}$ indicating its intrinsic complexity (i.e. the more complex $\mathcal{P}$ is, the smaller $\mathrm{Var}(\mathcal{P})$ is). This term is generally hard to compute and uninformative if $\mathcal{P}$ is not further specified. Therefore, we will take learning parity functions as an example to illustrate what concrete forms of CoT in this case help to improve the learnability.

**Illustrative Example: Parities.**   Boolean functions are mappings from an input space $\{\pm1\}^n$ of $n$ binary bits to an output space $\{\pm1\}$. Particularly, parities are a family of functions that compute the exclusive-or (XOR) of bits at some predefined positions in the input. The specific form of parity is determined by a subset $S \subseteq [n]$: the corresponding parity is defined as $\chi_S(x) = \prod_{i \in S} x[i]$. The distribution family is defined as $\mathcal{P} = \{(\chi_S(x), \mathcal{D}) : S \subseteq [n]\}$ where $\mathcal{D}$ is uniform over $\{\pm1\}^n$.

Parities are notoriously hard to learn (Kearns, 1998; Shalev-Shwartz et al., 2017; Daniely & Malach, 2020). Without CoT, using the proposed ICL algorithm (where it reduces to simple linear/kernel regression), the error has lower bound (12) with values $|\mathcal{H}_1'| = 1$ and $\mathrm{Var}(\mathcal{P}) = \mathcal{O}(2^{-n})$ (Malach & Shalev-Shwartz, 2022), which is exponentially bad w.r.t. the input size $n$. In contrast, with CoT, suppose we decompose the target function into

$$\chi_{1,S}(x)[i] = \begin{cases} x[i] & \text{for } i \in S \\ 1 & \text{for } i \notin S \end{cases}, \quad \chi_{2,S}(z) = \prod_i z[i], \tag{13}$$

where the first step $\chi_{1,S}(x)$ learns to select relevant features from $x$ while masking irrelevant ones, and the second step $\chi_{2,S}(z)$ computes XOR of all bits in $z$. Since each first step $\chi_{1,S}(x)$ is unique, with sufficient CoT examples generated with this decomposition, the ICL algorithm also returns unique $h_1$ with high probability, resulting in $|\mathcal{H}_1'| = 2^n$; this improves the lower bound by counteracting $\mathrm{Var}(\mathcal{P})$. In fact, one can easily show that the approximation error (i.e. the lower bound) can become zero through construction of $h$ that resembles $\chi_S(x)$. Regarding the upper bound (10), since both steps are learnable by a linear function class on fixed features, the error can be arbitrarily small, thus providing a guarantee that ICL algorithm can perform better with this specific CoT.

We would like to note here that the guarantee only holds for the proposed ICL algorithm rather than real-world LLMs. In fact, there is no guarantee that black-box Transformers will perform better with certain inputs, as there are always cases where they may produce bad results. Nevertheless, Section 5 will show that the predictions we derived here are highly consistent with the performance of LLMs.

# 5. Empirical Verification

## 5.1. Increasingly Complex Functions

To verify our results, we consider new arithmetic reasoning tasks and evaluate the performance of real-world LLMs, including the state-of-the-art GPT-4o and the less powerful GPT-3.5-turbo. This subsection studies the connection between the hardest step and the overall reasoning performance of LLMs. Since there lacks a testbed where one can control the hardness of the task and steps in the CoT, we propose a method to construct such tasks by incrementally building challenging tasks.

**Constructing Highly Challenging Tasks.**   Let us consider a class of elementary functions $\mathcal{F}_e$ where each function maps from the input space $\mathcal{X}$ to itself. In general, these elementary functions should be considered equally easy to learn. Then, we sample a sequence of these functions $f_1, f_2, \ldots, f_T \sim \mathcal{F}_e$; composing them gives us a target function $f = f_T \circ \cdots \circ f_2 \circ f_1 : \mathcal{X} \to \mathcal{X}$ whose complexity increases as $T$ increases. We consider an instantiation by defining the input space as the space of two integers $x \in \mathbb{Z}^2$. The elementary functions are defined as choosing one integer and using it to perform a basic arithmetic operation (drawn from $+$, $-$ or $\times$) with another number. Therefore, $\mathcal{F}_e$ consists of

$$\begin{aligned} z[0] &\leftarrow z[0] + z[1], & z[1] &\leftarrow z[1] + z[0] \\ z[0] &\leftarrow z[0] - z[1], & z[1] &\leftarrow z[1] - z[0] \\ z[0] &\leftarrow z[0] \times z[1], & z[1] &\leftarrow z[1] \times z[0]. \end{aligned} \tag{14}$$

While each elementary function in (14) is simple, the overall target function $f$ can become highly complex, possibly representing polynomial functions on $z[0]$ and $z[1]$ up to an arbitrary order and number of terms. Moreover, to quantify the hardest step, we do not reveal all intermediate results of $f$ in the demonstrations provided to LLMs. Instead, we stipulate that there exists at least one step $i \in [k]$ where the function from $z_{i-1}$ to $z_i$ is constructed from $H$ elementary functions, whereas all other steps use fewer of them. For example, given $H = 3$ elementary functions $f_1 : z[0] \leftarrow z[0] + z[1]$, $f_2 : z[1] \leftarrow z[1] \times z[0]$ and $f_3 : z[1] \leftarrow z[1] - z[0]$, the hardest step can be expressed as $f_3 \circ f_2 \circ f_1$ :

$$\begin{cases} z_i[0] &= z_{i-1}[0] + z_{i-1}[1] \\ z_i[1] &= (z_{i-1}[0] + z_{i-1}[1])(z_{i-1}[1] - 1) \end{cases} \tag{15}$$

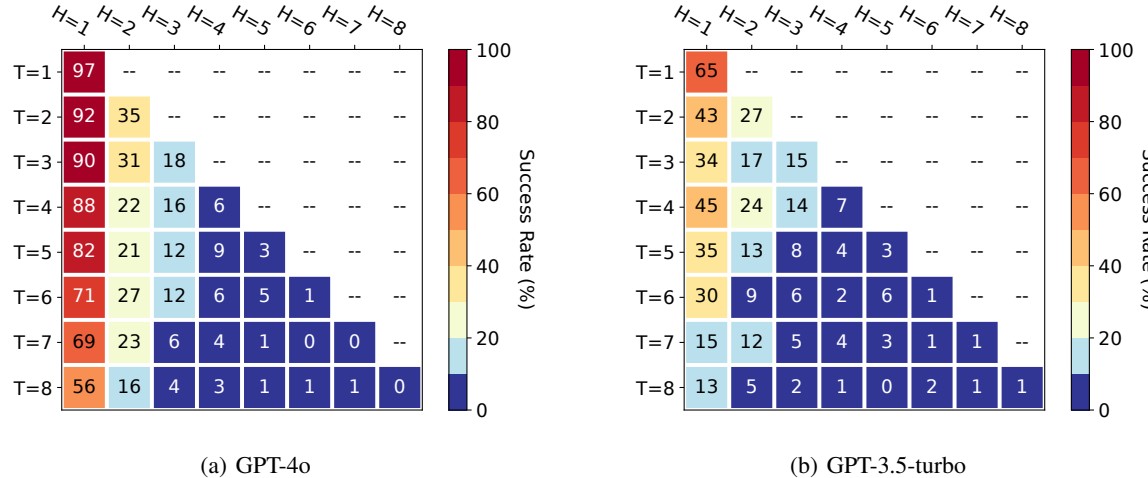

(a) GPT-4o

(b) GPT-3.5-turbo

Figure 1: Success rate of GPT-4o and GPT-3.5-turbo for learning compositional functions. $T$ denotes the number of elementary functions used to construct the target function; $H$ denotes the maximal number of elementary functions to construct a reasoning step.

See more detailed experimental setup in Appendix B.1.

**Results.** We test the performance of real-world LLMs on the reasoning task described above with respect to different overall hardness $T$ and the hardest step $H$. We report their success rates across 100 i.i.d. sampled target functions for each $T$ and $H$. And for each target function, the LLMs are provided with 10 demonstrations and asked to infer the computation process and apply it to derive the output for an unseen input. As shown in Fig. 1 (and more results in Appendix B.1), the success rate of LLMs quickly drops as $H$ increases. In particular, GPT-4o can successfully learn the target function in most cases when $H = 1$; however, it performs significantly worse as $H$ increases from 1 to 4, then fails as $H$ becomes even larger. These phenomena corroborate our result that reducing the complexity of the hardest step is critical for LLMs to successfully handle the task.

### 5.2. Canonical Boolean Functions

Next, we test whether the specific forms of CoT obtained from our learning theoretic analysis aligns with the performance of real-world LLMs. In particular, we evaluate LLMs on two families of Boolean functions: parities and disjunctive normal form (DNF), which are known hard to learn in theory (Malach & Shalev-Shwartz, 2022).

**Task Descriptions.** An $(n, k)$-parity function computes the XOR ($\oplus$) of a subset of $k$ variables from a total of $n$ input binary bits ($n$ is 10 in experiments). It outputs 1 if an odd number of the $k$ relevant variables are 1, and 0 otherwise. A DNF function is a disjunction (logical OR) of conjunctions

(logical ANDs) of literals; and in the experiments, we consider a family of 3-term DNFs $f(x) = \vee_{i=1}^{3} \wedge_{j=1}^{w} (x_{ij} \vee m_{ij})$ where $w$ is the width and $m \in \{\pm 1\}^{3w}$ is a latent variable whose value determines the target function (i.e. $m_{ij} = 1$ invalidates $x_{ij}$). More details in Appendix B.2.

**Results.** Similarly, for each $k$ in parities and $w$ in DNFs, we i.i.d. sample 100 target functions.[3] For each function, we provide LLMs with 100 in-context examples, ask them to find patterns in them and return the output for a query. Table 1 and 2 report their success rate. Particularly, in terms of parities, we find even the SOTA model GPT-4o generally performs no better than random guessing (with an expected accuracy of 50%) when $k > 3$.

Then, we provide LLMs CoT examples with intermediate results that are provably effective by applying our results derived in Section 4: for parities, the intermediate result is defined as $z[i] = x[i]$ if $i \in S$ otherwise 0; for DNFs, $z[i, j] = x[i, j]$ if $m[i, j] = 0$ otherwise 1. Results in table 1 and 2 clearly demonstrate the designed CoT significantly improves the performance, e.g. GPT-4o achieves an almost perfect success rate of 94 on $(10, 4)$-parity, while without CoT the success rate is 54, which is almost random.

## 6. Related Work

In-context learning (ICL) (Garg et al., 2022) has gained significant theoretical interest since its introduction (Brown et al., 2020). Recent works (Akyürek et al., 2023; Von Os-

---

[3]For parities, we sample from a uniform distribution; for DNF, we sample from a non-uniform distribution to ensure the label (0/1) is balanced for $w \geq 3$.

| $(n,k)$-Parities | (10,1) | (10,2) | (10,3) | (10,4) | (10,5) | (10,6) | (10,7) | (10,8) | (10,9) | (10,10) |
|---|---|---|---|---|---|---|---|---|---|---|
| GPT-4o w/o CoT | 87 | 69 | 63 | 54 | 51 | 48 | 50 | 52 | 47 | 51 |
| GPT-4o w CoT | **92** | **95** | **97** | **94** | **87** | **73** | **66** | 58 | **62** | 50 |
| GPT-3.5-turbo w/o CoT | 75 | 62 | 60 | 47 | 59 | 58 | 51 | 56 | 55 | 54 |
| GPT-3.5-turbo w CoT | 80 | 76 | 72 | 74 | 75 | 69 | 57 | **63** | 57 | **57** |

Table 1: Success rate (%) of GPT-4o and GPT-3.5-turbo of learning (n,k)-parities.

| 3-Term DNF | Width 3 | Width 4 | Width 5 | Width 6 | Width 7 | Width 8 | Width 9 | Width 10 |
|---|---|---|---|---|---|---|---|---|
| GPT-4o w/o CoT | 85 | 81 | 77 | 73 | 68 | 66 | 62 | 74 |
| GPT-4o w CoT | **96** | **87** | 86 | **88** | **81** | **80** | **80** | **84** |
| GPT-3.5-turbo w/o CoT | 74 | 68 | 67 | 62 | 55 | 58 | 64 | 53 |
| GPT-3.5-turbo w CoT | 90 | 78 | **87** | 81 | 65 | 73 | 70 | 73 |

Table 2: Success rate (%) of GPT-4o and GPT-3.5-turbo of learning 3-term DNF.

wald et al., 2023; Dai et al., 2023; von Oswald et al., 2023; Cheng et al., 2024; Li et al., 2023a) explain ICL as performing certain optimization algorithms by showing parameter configurations that enable Transformers to implement gradient descent (Von Oswald et al., 2023) or its variants (Giannou et al., 2024; Fu et al., 2023a) to learn linear models. Some studies further show that a single-layer Transformer converges to weights that align with these constructions when pre-trained on ICL tasks (Ahn et al., 2023; Mahankali et al., 2023; Zhang et al., 2023). Notably, Cheng et al. (2024) demonstrated that Transformers could implement kernel gradient descent; Bhattamishra et al. (2023) empirically studied various models' capability to in-context learn discrete functions. However, what class of ICL algorithms Transformers can implement with CoT remains open and is investigated in this paper.

Chain-of-thought (CoT) prompting (Wei et al., 2022; Reynolds & McDonell, 2021; Nye et al., 2021) augments demonstrations with intermediate reasoning steps. Despite its effectiveness in various reasoning tasks (Kojima et al., 2022; Yao et al., 2024; Lanchantin et al., 2024), theoretical analyses are scarce and mostly focus on the expressiveness perspective (Li et al., 2024; Feng et al., 2024). The most closely related work is by Li et al. (2023b), where the authors show that Transformers can first filter inputs and then perform linear regression to learn MLPs (specifically with Leaky-ReLU). We extend existing work by showing that Transformers can learn a richer family of compositional functions. The generalization analyses further provide practical lessons on designing CoT prompts.

## 7. Limitations, Discussion, Broader Impacts

We would like to note a limitation that Section 4 are specifically derived based on the ICL algorithm presented in Section 3 rather than real-world LLMs. It is uncertain whether real-world LLMs actually implement this ICL algorithm, and it is also infeasible to verify due to their black-box nature. Without precise knowledge of the computations or limits of pre-trained Transformers, providing guarantees for their performance on various prompts is prohibitively difficult. Nevertheless, we demonstrate the feasibility of doing so by focusing on specific ICL algorithms that can be expressed by Transformers, with predictions aligning well with practice. The emergence of simplified variants of these algorithms are supported both experimentally and theoretically (Von Oswald et al., 2023; Cheng et al., 2024; Ahn et al., 2023; Mahankali et al., 2023; Zhang et al., 2023). Another limitation is regarding the tokenization and embedding of prompts, which remain open questions without an agreed standard. Our setup follows previous works, and results could potentially be adapted to other settings.

Therefore, an interesting direction for future work is to understand the pre-training of Transformers and the exact ICL mechanism that emerges during this process. Extending our analysis to explore other variants of the ICL algorithm with CoT, simpler constructions, and the role of intermediate steps in more reasoning tasks would also be valuable. We believe this paper can promote understanding of the underlying mechanism of CoT and has positive impacts by providing high-level guidance for designing prompts. Given the theoretical nature of this paper, we do not foresee any immediate negative societal impacts.

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

# A. Proofs

## A.1. Theorem 3.1: Construction

Given demonstrations $\{(z_0^{(j)}, z_1^{(j)}, \cdots, z_{k-1}^{(j)}, z_k^{(j)})\}_{j=1}^N$, we could create $k$ training sets, each of which defines a loss function quantifying the error of a particular predictor $h_i$. These loss functions are

$$\left\{ \mathcal{L}_i = \frac{1}{2} \sum_{j=1}^N \left\| h_i(z_{i-1}^{(j)}) - z_i^{(j)} \right\|_2^2 : i \in [k] \right\}. \tag{16}$$

Note that while the overall predictor

$$h = h_k \circ \cdots \circ h_2 \circ h_1 = W_k \phi_k(\cdots (W_2 \phi_2(W_1 \phi_1(x)))) \tag{17}$$

is a non-linear function, loss functions in (16) are convex with respective to the weights of their corresponding linear predictors $\{W_i : i \in [k]\}$. Let us also denote $h_{\leq k'} = h_{k'} \circ \cdots \circ h_2 \circ h_1$ for $k' \leq k$. Using gradient descent to minimize $\mathcal{L}_i$ with fixed step size $\eta$ induces the following training dynamics in weight space

$$W_i \leftarrow W_i - \eta \nabla_{W_i} \mathcal{L}_i = W_i + \eta \left( Z_i - h_i(Z_{i-1}) \right) \phi_i(Z_{i-1})^\top \tag{18}$$

where $h_i(Z_{i-1}) = [h_i(z_{i-1}^{(j)})]_{j \in [N]} \in \mathbb{R}^{d(\mathcal{Z}_i) \times N}$, $\phi_i(Z_{i-1}) = [\phi_i(z_{i-1}^{(j)})]_{j \in [N]} \in \mathbb{R}^{K \times N}$. Note that difference between our setup and the conventional supervised learning setup is that, the latter is interested in the variation of output with different weights, while in this paper, we are also interested in the dynamics of intermediate results. Particularly:

- For $i' < i$, the dynamics of intermediate results induced by GD is

$$h_{\leq i'}(x^{(N+1)}) \leftarrow h_{\leq i'}(x^{(N+1)}), \tag{19}$$

namely the variation of upper layer weights does not affect lower layer representations (i.e. intermediate results).

- For $i' = i$, the dynamics is

$$
\begin{aligned}
h_{\leq i}(x^{(N+1)}) &= W_i \phi_i(h_{\leq i-1}(x^{(N+1)})) \\
&\leftarrow (W_i - \eta \nabla_{W_i} \mathcal{L}_i) \, \phi_i(h_{\leq i-1}(x^{(N+1)})) \\
&= h_{\leq i}(x^{(N+1)}) + \eta \left( Z_i - h_i(Z_{i-1}) \right) \phi_i(Z_{i-1})^\top \phi_i(h_{\leq i-1}(x^{(N+1)})).
\end{aligned}
\tag{20}
$$

Let $\kappa_i$ be the kernel function defined by the feature map $\phi_i$, we have

$$h_{\leq i}(x^{(N+1)}) \leftarrow h_{\leq i}(x^{(N+1)}) + \eta \left( Z_i - h_i(Z_{i-1}) \right) \kappa_i(Z_{i-1}, h_{\leq i-1}(x^{(N+1)})). \tag{21}$$

- For $i' > i$, the dynamics is

$$
\begin{aligned}
h_{\leq i'}(x^{(N+1)}) &= h_i \circ \cdots \circ h_{i'+1} \circ h_{\leq i'}(x^{(N+1)}) \\
&\leftarrow h_i \circ \cdots \circ h_{i'+1} \left( h_{\leq i}(x^{(N+1)}) + \eta \left( Z_i - h_i(Z_{i-1}) \right) \kappa_i(Z_{i-1}, h_{\leq i-1}(x^{(N+1)})) \right)
\end{aligned}
\tag{22}
$$

which in general intractable since $h_i \circ \cdots \circ h_{i'+1}$ is non-linear. However, if upper layer weights in $h_i, \cdots, h_{i'+1}$ are 0, $h_{\leq i'}(x^{(N+1)})$ will become 0 as well and thus we can circumvent (22).

Recall also that based on the definition in Section 2, the self-attention layer can be written as

$$\text{(Self-Attention)} \qquad e^{(N+1)} \leftarrow e^{(N+1)} + W_V E \sigma \left( E^\top W_K^\top W_Q e^{(N+1)} \right). \tag{23}$$

where $e = (x, z_1, \cdots, z_{k-1}, z_k) \in \mathbb{R}^d$ at the input layer, $d = \sum_{i=0}^k d(\mathcal{Z}_i)$.

In the following construction, we show that in the forward pass of Transformer, (23) could express dynamics of all intermediate results, including (19), (21) and (22), based on a certain order in which minimization of losses in (16) is performed. Particularly, in our construction, Transformer will sequentially minimize loss functions in (16). In other words,

the lower layers of the Transformer learn prior reasoning steps, while upper layers of the Transformer learn later reasoning steps.

We begin by constructing a projection matrix $P$ that projects $e$ onto an expanded space

$$
P = \begin{pmatrix}
I_{d(\mathcal{Z}_0)} & 0 & \cdots & \cdots & 0 \\
0 & I_{d(\mathcal{Z}_1)} & \cdots & \cdots & 0 \\
0 & I_{d(\mathcal{Z}_1)} & \cdots & \cdots & 0 \\
0 & 0 & I_{d(\mathcal{Z}_2)} & \cdots & 0 \\
0 & 0 & I_{d(\mathcal{Z}_2)} & \cdots & 0 \\
\vdots & \vdots & \vdots & \ddots & \vdots \\
0 & 0 & 0 & \cdots & I_{d(\mathcal{Z}_k)}
\end{pmatrix}
\tag{24}
$$

which copies certain dimensions in the initial embedding and transforms it into

$$
e = (x, z_1 - h_1(x), z_1, \cdots, z_{k-1} - h_{k-1}(z_{k-2}), z_{k-1}, z_k - h_k(z_{k-1})) \in \mathbb{R}^{d'}
\tag{25}
$$

where $d' = 2d - d(\mathcal{X}) - d(\mathcal{Z}_k)$ and $h_i(z_{i-1}) = 0$ for $i \in [k]$ at initialization. For the query input, that is defined as

$$
e^{(N+1)} = \left( x^{(N+1)}, -h_1(x^{(N+1)}), 0, \cdots, -h_{\leq k-1}(x^{(N+1)}), 0, -h_{\leq k}(x^{(N+1)}) \right) \in \mathbb{R}^{d'}
\tag{26}
$$

where $h_{\leq i}(x^{(N+1)}) = 0$ at initialization, $h_{\leq k}$ is exactly the final prediction $h$ we desire.

For layers that minimize the $i$-th step's loss function $\mathcal{L}_i$, we construct Transformer weights in the corresponding self-attention layer as:

$$
W_V = \begin{pmatrix}
0_{d_l(i)} & 0 & 0 \\
0 & -\eta I_{d(\mathcal{Z}_i)} & 0 \\
0 & 0 & 0_{d_r(i)}
\end{pmatrix}
\tag{27}
$$

where $d_l(i) = 2 \sum_{j=0}^{i-1} d(\mathcal{Z}_j) - d(\mathcal{X})$, $d_r(i) = 2 \sum_{j=i}^{k} d(\mathcal{Z}_j) - d(\mathcal{Z}_k) - d(\mathcal{Z}_i)$, selecting residuals in the embedding.

$$
W_K^\top W_Q = \begin{pmatrix}
0_{d'_l(i)} & 0 & 0 \\
0 & I_{d(\mathcal{Z}_{i-1})} & 0 \\
0 & 0 & 0_{d'_r(i)}
\end{pmatrix}
\tag{28}
$$

where $d'_l(i) = 2 \sum_{j=0}^{i-1} d(\mathcal{Z}_j) - d(\mathcal{X}) - d(\mathcal{Z}_{i-1})$, $d'_r(i) = 2 \sum_{j=i}^{k} d(\mathcal{Z}_j) - d(\mathcal{Z}_k)$, selecting intermediate results (or inputs) for computing the attention matrix. Stacking $t$ self-attention layers minimizes the loss function for $t$ steps. For $i \in [k]$, apply this procedure to sequentially minimize loss functions in (16) gives us the desired result. The output includes prediction for all intermediate steps and the final prediction.

It is not hard to verify linear/kernel regression in Von Oswald et al. (2023); Cheng et al. (2024) are special cases of our construction.

### A.2. Lemma 4.1: Upper Bound

Given a target function $f$ and an input distribution $\mathcal{D}(x)$, the ICL algorithm $A$ returns a predictor $h$ based on $N$ i.i.d. samples, whose expected error is defined as $\mathbb{E}_{x \sim \mathcal{D}(x)}[l(h(x), f(x))]$. For a family of distributions $\mathcal{P} \ni (f, \mathcal{D})$, we define the error as

$$
\Delta(\mathcal{P}, h) \triangleq \max_{(f, \mathcal{D}) \in \mathcal{P}} \mathbb{E}_{x \sim \mathcal{D}(x)} \left[ l\left( h(x), f(x) \right) \right],
\tag{29}
$$

where we slightly abuse notation here as $h$ is also dependent on the distribution $(f, \mathcal{D})$ and the learning algorithm. For a certain decomposition operator $T$, the target function can be expressed as $f = f_2 \circ f_1$ and the predictor $h = h_2 \circ h_1$. We have

$$
\Delta(\mathcal{P}, h) = \max_{(f, \mathcal{D}) \in \mathcal{P}} \mathbb{E}_{x \sim \mathcal{D}(x)} \left[ \frac{1}{2} \left( (h_2 \circ h_1)(x) - (h_2 \circ f_1)(x) + (h_2 \circ f_1)(x) - f(x) \right)^2 \right]
\tag{30}
$$

Suppose the feature map $\phi(x)$ for $h_2$ has Lipschitz constant

$$\text{Lip}(\phi) = \sup_{x \neq x'} \frac{\|\phi(x) - \phi(x')\|_2}{\|x - x'\|_2}, \tag{31}$$

and by Jensen's inequality, we have

$$\Delta(\mathcal{P}, h) \leq \max_{(f,\mathcal{D}) \in \mathcal{P}} \mathbb{E}_{x \sim \mathcal{D}(x)} \left[ ((h_2 \circ h_1)(x) - (h_2 \circ f_1)(x))^2 + ((h_2 \circ f_1)(x) - f(x))^2 \right] \tag{32}$$

$$\leq \max_{(f,\mathcal{D}) \in \mathcal{P}} \mathbb{E}_{x \sim \mathcal{D}(x)} \left[ B^2 \text{Lip}(\phi)^2 \|h_1(x) - f_1(x)\|_2^2 + (h_2(z) - f_2(z))^2 \right] \tag{33}$$

$$\leq c_{B,\phi} \max_{(f,\mathcal{D}) \in \mathcal{P}} \mathbb{E}_{x \sim \mathcal{D}(x)} \left[ \|h_1(x) - f_1(x)\|_2^2 + (h_2(z) - f_2(z))^2 \right] \tag{34}$$

where $c_{B,\phi} = \max\{1, B^2 \text{Lip}(\phi)^2\}$ is a constant determined by the definition of hypothesis class. Given decomposition operator $T$, the distribution family can be decomposed into $\mathcal{P}_{T,1}$ and $\mathcal{P}_{T,2}$. It follows that

$$\frac{\Delta(\mathcal{P}, h)}{2c_{B,\phi}} \leq \max_{(f,\mathcal{D}) \in \mathcal{P}} \mathbb{E}_{x \sim \mathcal{D}(x)} \left[ \frac{1}{2} \|h_1(x) - f_1(x)\|_2^2 \right] + \max_{(f,\mathcal{D}) \in \mathcal{P}} \mathbb{E}_{x \sim \mathcal{D}(x)} \left[ \frac{1}{2} (h_2(z) - f_2(z))^2 \right] \tag{35}$$

$$= \max_{(f_1,\mathcal{D}) \in \mathcal{P}_{T,1}} \mathbb{E}_{x \sim \mathcal{D}(x)} \left[ l(h_1(x), f_1(x)) \right] + \max_{(f_2,\mathcal{D}') \in \mathcal{P}_{T,2}} \mathbb{E}_{z \sim \mathcal{D}(z)} \left[ l(h_2(z), f_2(z)) \right] \tag{36}$$

$$= \Delta(\mathcal{P}_{T,1}, h_1) + \Delta(\mathcal{P}_{T,2}, h_2) \tag{37}$$

Thus we get the desired upper bound

$$\Delta(\mathcal{P}, h) \leq 4c_{B,\phi} \max\{\Delta(\mathcal{P}_{T,1}, h_1), \Delta(\mathcal{P}_{T,2}, h_2)\}. \tag{38}$$

### A.3. Theorem 4.3: Lower Bound

The in-context learning error has approximation error lower bound, that is the minimum error achievable by a predictor in the hypothesis class $\mathcal{H} = \{h_2 \circ h_1 : h_1 \in \mathcal{H}'_1, h_2 \in \mathcal{H}_2\}$

$$\max_{(f,\mathcal{D}) \in \mathcal{P}} \mathbb{E}_{x \sim \mathcal{D}(x)} \left[ l(h(x), f(x)) \right] \geq \max_{(f,\mathcal{D}) \in \mathcal{P}} \min_{(h_1,h_2) \in \mathcal{H}'_1 \times \mathcal{H}_2} \mathbb{E}_{x \sim \mathcal{D}(x)} \left[ l(h(x), f(x)) \right]. \tag{39}$$

Thus it suffices to lower bound the approximation error.

To do so, notice that the learned first-step predictor $h_1$ is from finite function class $\mathcal{H}'_1$, which is a subset of the initial hypothesis class $\mathcal{H}_1$. We show the approximation power of $h(x)$ with finite-sized $\mathcal{H}'_1$ is lower bounded by a linear class whose size depends on $\mathcal{H}'_1$. In particular, suppose the hypothesis class is

$$\mathcal{H}'_1 = \{h_{1,1}, h_{1,2}, \cdots, h_{1,|\mathcal{H}'_1|}\}, \tag{40}$$

and based on the index of $h_1$ in $\mathcal{H}'_1$, the predictor $h(x)$ can be re-written as

$$h(x, j) \triangleq W_2 (\phi \circ h_{1,j})(x) = \sum_{i=1}^{K} W_{2,i} \phi_{i,j}(x) \tag{41}$$

$$= \sum_{i=1}^{K} \sum_{i'=1}^{|\mathcal{H}'_1|} U_{i,i'} \phi_{i,i'}(x) \tag{42}$$

where $\phi_{i,j} = \phi_i \circ h_{1,j}$ and $U_{i,i'} = W_{2,i}$ if $i' = j$ otherwise 0. As (42) is an inner product of weight vector $U \in \mathbb{R}^{K|\mathcal{H}'_1|}$ and feature vector $\phi(x) \in \{\pm 1\}^{K|\mathcal{H}'_1|}$ in an expanded space, $h(x, j)$ reduces to a linear function.

Since the squared loss $l(h(x, j), f(x))$ is convex w.r.t. $U$ for arbitrary $x$ and $j$, its expectation $L_{f,\mathcal{D}}(h) \triangleq \mathbb{E}_{x \sim \mathcal{D}(x)} [l(h(x, j), f(x))]$ is also convex w.r.t. $U$. Moreover, $h(x, j) = 0$ when $W_2$ or $U$ goes to 0. Therefore, given any $(f, \mathcal{D}) \in \mathcal{P}$ and any predictor $h$ with fixed $W_2$ and $j$ (and thus fixed $U$), by first-order condition, we have

$$L_{f,\mathcal{D}}(h) \geq L_{f,\mathcal{D}}(0) + \langle U - 0, \nabla_U L_{f,\mathcal{D}}(h)|_{U=0} \rangle \tag{43}$$

$$\geq \frac{1}{2} - \|U\|_2 \|\nabla_U L_{f,\mathcal{D}}(h)|_{U=0}\|_2 \tag{44}$$

where the last equation uses the fact $L_{f,\mathcal{D}}(0) = \mathbb{E}_{x \sim \mathcal{D}(x)}\left[l\left(0, f(x)\right)\right] = \frac{1}{2}$ and inequality $\langle v, u \rangle \geq -\|u\|_2 \|v\|_2$.

Notice $U$ has the same norm as $W_2$ and thus $\|U\|_2 \leq B$. Moreover, we have

$$\nabla(f, \mathcal{D}) \triangleq \left\|\nabla_U L_{f,\mathcal{D}}(h)|_{U=0}\right\|_2^2 = \left\|\nabla_U \mathbb{E}_{x \sim \mathcal{D}(x)}\left[l\left(h(x, j), f(x)\right)\right]|_{U=0}\right\|_2^2 \tag{45}$$

$$= \left\|\mathbb{E}_{x \sim \mathcal{D}(x)}\left[\nabla_U l\left(h(x, j), f(x)\right)|_{U=0}\right]\right\|_2^2 \tag{46}$$

$$= \left\|\mathbb{E}_{x \sim \mathcal{D}(x)}\left[\nabla_U \frac{1}{2}(\langle U, \phi(x)\rangle - f(x))^2|_{U=0}\right]\right\|_2^2 \tag{47}$$

$$= \left\|\mathbb{E}_{x \sim \mathcal{D}(x)}\left[\phi(x) f(x)\right]\right\|_2^2 \tag{48}$$

$$= \sum_{i=1}^{K} \sum_{j=1}^{|\mathcal{H}_1'|} \mathbb{E}_{x \sim \mathcal{D}(x)}\left[\phi_{i,j}(x) f(x)\right]^2 \tag{49}$$

Subjecting it to (44) gives us that, for any $(f, \mathcal{D}) \in \mathcal{P}$ and any predictor $h(x)$ obtained from in-context learning, i.e.

$$\min_{(h_1, h_2) \in \mathcal{H}_1' \times \mathcal{H}_2} L_{f,\mathcal{D}}(h) \geq \frac{1}{2} - B \cdot \nabla(f, \mathcal{D})^{\frac{1}{2}} \tag{50}$$

Following from (39), the in-context learning error has lower bound

$$\Delta(\mathcal{P}, h) \geq \max_{(f, \mathcal{D}) \in \mathcal{P}} \min_{(h_1, h_2) \in \mathcal{H}_1' \times \mathcal{H}_2} L_{f,\mathcal{D}}(h) \tag{51}$$

$$\geq \max_{(f, \mathcal{D}) \in \mathcal{P}} \left[\frac{1}{2} - B \cdot \nabla(f, \mathcal{D})^{\frac{1}{2}}\right] \tag{52}$$

$$\geq \mathbb{E}_{(f, \mathcal{D}) \in \mathcal{P}} \left[\frac{1}{2} - B \cdot \nabla(f, \mathcal{D})^{\frac{1}{2}}\right] \tag{53}$$

$$= \frac{1}{2} - B \cdot \mathbb{E}_{(f, \mathcal{D}) \in \mathcal{P}} \left[\nabla(f, \mathcal{D})^{\frac{1}{2}}\right] \tag{54}$$

By noting $\mathbb{E}[X^{\frac{1}{2}}] \leq \mathbb{E}[X]^{\frac{1}{2}}$, we have

$$\Delta(\mathcal{P}, h) \geq \frac{1}{2} - B \cdot \mathbb{E}_{(f, \mathcal{D}) \in \mathcal{P}} \left[\nabla(f, \mathcal{D})\right]^{\frac{1}{2}} \tag{55}$$

$$= \frac{1}{2} - B \cdot \mathbb{E}_{(f, \mathcal{D}) \in \mathcal{P}} \left[\sum_{i=1}^{K} \sum_{j=1}^{|\mathcal{H}_1'|} \mathbb{E}_{x \sim \mathcal{D}(x)}\left[\phi_{i,j}(x) f(x)\right]^2\right]^{\frac{1}{2}} \tag{56}$$

Let

$$\mathrm{Var}(\mathcal{P}) \triangleq \sup_{\phi} \mathbb{E}_{(f, \mathcal{D}) \in \mathcal{P}} \left[\mathbb{E}_{x \sim \mathcal{D}(x)}\left[\phi(x) f(x)\right]^2\right], \tag{57}$$

which is determined by target functions and distributions in $\mathcal{P}$, and could be understood as the intrinsic complexity of the distribution family (i.e. the more complex $\mathcal{P}$ is, the smaller $\mathrm{Var}(\mathcal{P})$ is). It follows that,

$$\Delta(\mathcal{P}, h) \geq \frac{1}{2} - B \sqrt{\sum_{i=1}^{K} \sum_{j=1}^{|\mathcal{H}_1'|} \mathbb{E}_{(f, \mathcal{D}) \in \mathcal{P}} \left[\mathbb{E}_{x \sim \mathcal{D}(x)}\left[\phi_{i,j}(x) f(x)\right]^2\right]} \tag{58}$$

$$\geq \frac{1}{2} - B \sqrt{K |\mathcal{H}_1'| \, \mathrm{Var}(\mathcal{P})}, \tag{59}$$

completing the proof.

# B. Experimental Details and Additional Results

## B.1. Increasingly Complex Functions

**Sampling.**  For this task, we sample a sequence of functions from a set of elementary functions with the following probabilities:

$$P\left(z[0] \leftarrow z[0] + z[1]\right) = \frac{1}{8}, \quad P\left(z[0] \leftarrow z[0] - z[1]\right) = \frac{1}{8}, \quad P\left(z[0] \leftarrow z[0] \times z[1]\right) = \frac{1}{4},$$

$$P\left(z[1] \leftarrow z[1] + z[0]\right) = \frac{1}{8}, \quad P\left(z[1] \leftarrow z[1] - z[0]\right) = \frac{1}{8}, \quad P\left(z[1] \leftarrow z[1] \times z[0]\right) = \frac{1}{4}.$$

Here, the multiplication operation is more likely to be sampled than addition or subtraction. For the input values, we uniformly select two unique integers from the set $\{2, 3, \ldots, 10\}$. This range is chosen to avoid excessively large numbers, which are difficult to handle, and trivial cases, such as when $x[0] = x[1]$, which could result in zero values that are easily predictable by LLMs. For this task, we explicitly ensure that each sample is unique to prevent repeated samples in training and testing. This is done to avoid scenarios where LLMs could simply memorize the results from the demonstrations and use them to answer the query.

**CoT Prompting.**  Given a certain $H$, namely the maximal number of elementary functions to construct a reasoning step, we implement it by randomly masking $H - 1$ consecutive intermediate results. For example, when $T = 6$ and $H = 3$, an example prompt is given as follows:

```
Given two numbers, sequentially apply predefined arithmetic operations (addition,
↪    subtraction, multiplication) to transform them. Each step involves a specific
↪    predefined operation on one of the numbers. If any operations or intermediate
↪    results are missing, deduce these to complete the transformation and arrive
↪    at the final output.

Input: 7, 5
Step1: 7, -2
Step2: 5, -2
Step3: missing
Step4: missing
Step5: 5, 75
Output: -70, 75

Input: 2, 3
Step1: 2, 1
Step2: 3, 1
Step3: missing
Step4: missing
Step5: 3, 36
Output: -33, 36

...

Input: 5, 8
What is the output? Your answer should end in the format 'Step1: ?, Step2: ?,
↪    ..., Output:?'.
```

Note that the missing steps are consistent for one trial. We test 100 times to compute the success rate.

**Additional Results.**  In addition to reporting the success rate of LLMs for predicting the final output as presented in Section 5, we also evaluate their success rate for predicting intermediate results. This provides a more comprehensive assessment of the LLMs' performance, as even they might fail to predict the final step but could still succeed in predicting the intermediate steps.

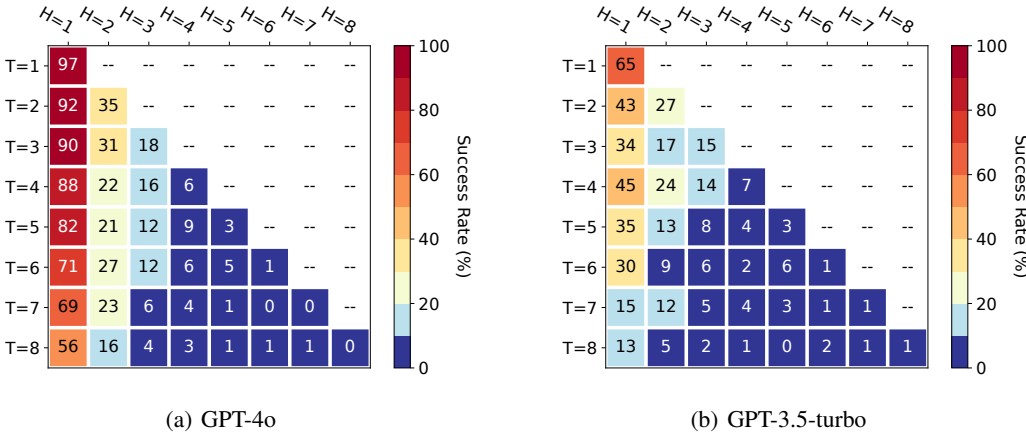

Figure 2: Success rate for predicting the last step (i.e. namely the output).

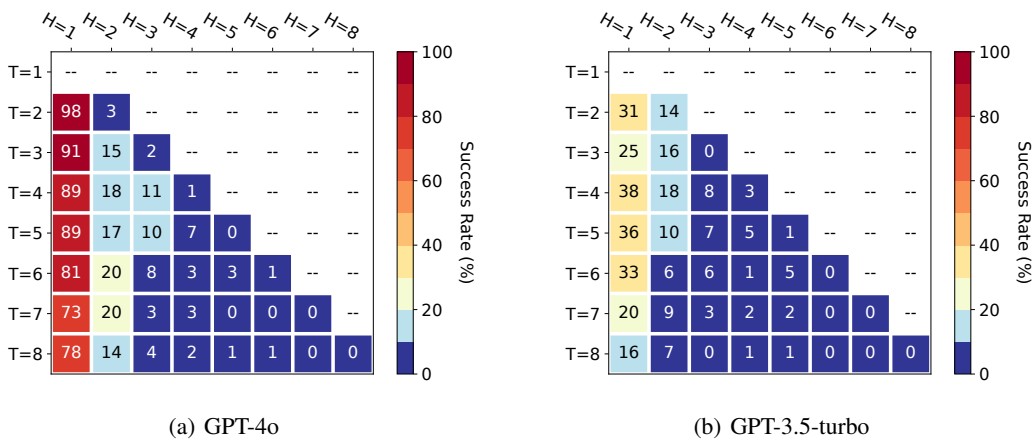

Figure 3: Success rate for predicting the second to last step.

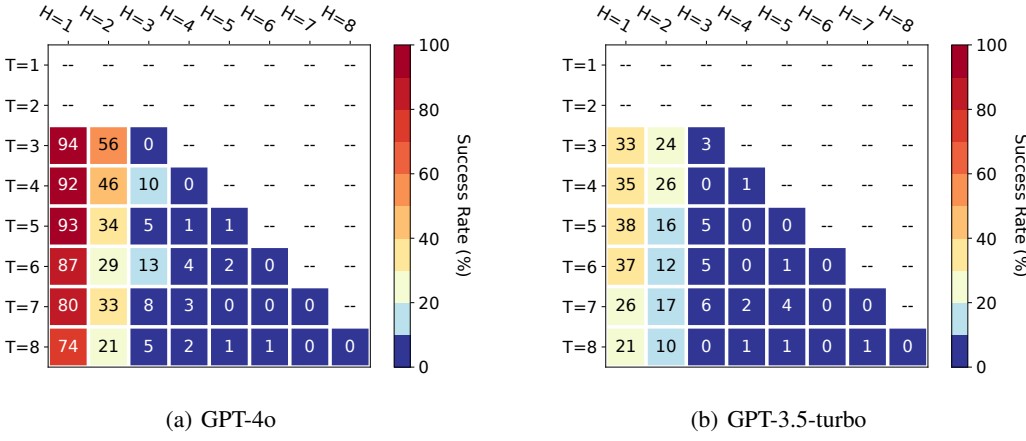

Figure 4: Success rate for predicting the third to last step.

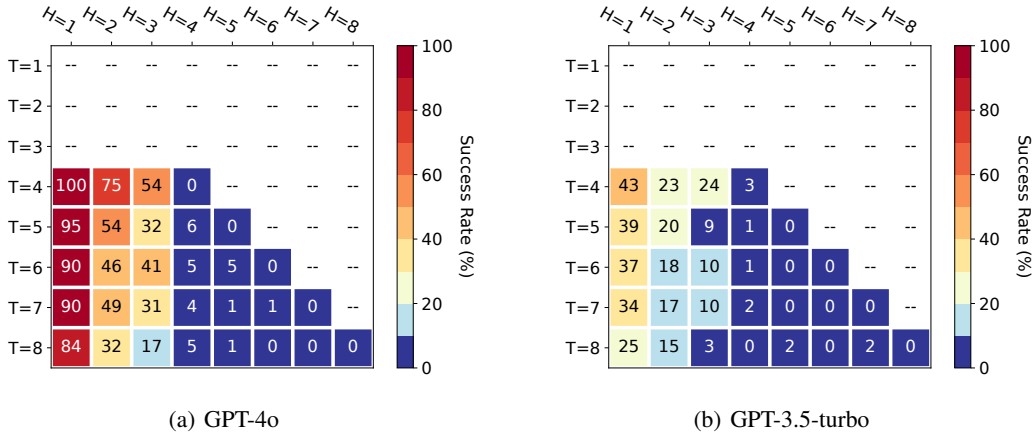

(a) GPT-4o

(b) GPT-3.5-turbo

Figure 5: Success rate for predicting the fourth to last step.

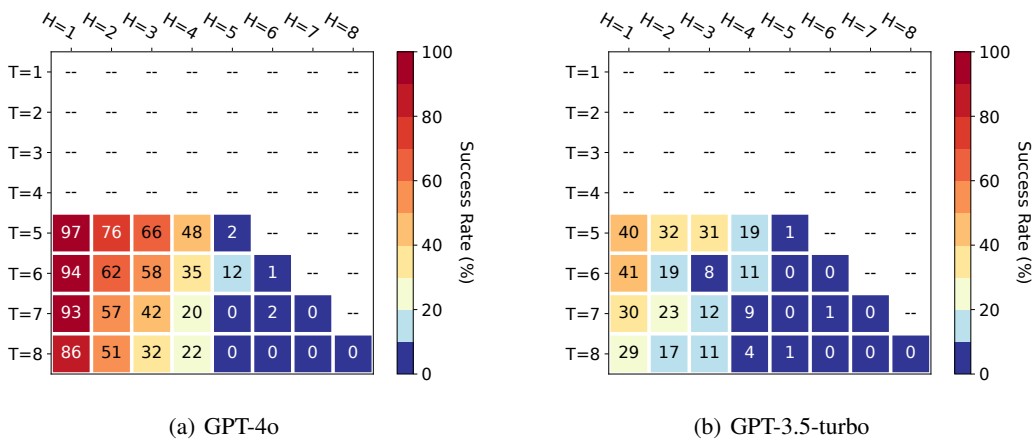

(a) GPT-4o

(b) GPT-3.5-turbo

Figure 6: Success rate for predicting the fifth to last step.

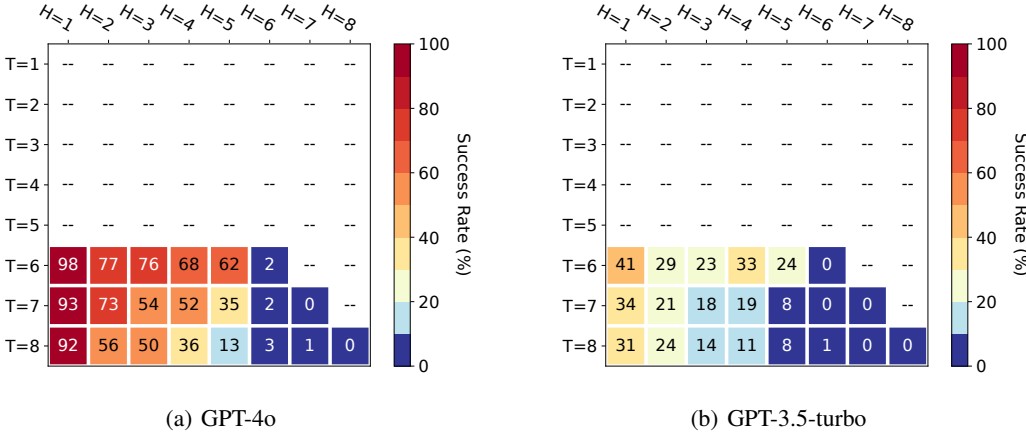

(a) GPT-4o

(b) GPT-3.5-turbo

Figure 7: Success rate for predicting the sixth to last step.

**B.2. Canonical Boolean Functions**

Examples of standard and CoT prompts for $(10, 4)$-parity and DNF with width 6 are given as follows:

- Standard prompt for parity:

```
Predict the output based on a pattern in the input binary string.

Input: 1010101100
Output: 0

Input: 0100000011
Output: 0

Input: 0110010000
Output: 1

Input: 1010100001
Output: 0

Input: 1010100001
Output: 0

...

Input: 1011000111
What is the output? Directly answer the question in the format 'Output:'.
```

- CoT prompt for parity:

```
Replace some bits located at specific predefined positions in the binary string
↪   with 0 to form a new string. Then, based on some patterns in the new string
↪   to predict the output.

Input: 1000010001
New string: 0000010001
Output: 0

Input: 0100110111
New string: 0000110001
Output: 1

Input: 0101001000
New string: 0001000000
Output: 1

Input: 0100000010
New string: 0000000000
Output: 0

Input: 1011010000
New string: 0001010000
Output: 0

...
```

```
Input: 0100100110
What is the output? Directly answer the question in the format 'New string:,
↪  Output:'.
```

- Standard prompt for DNF:

```
Predict the output based on a pattern in the input binary string.

Input: 001010 000001 010011
Output: 1

Input: 100111 011001 010111
Output: 1

Input: 010010 011010 101011
Output: 1

Input: 010101 001101 001001
Output: 1

Input: 111110 011001 010111
Output: 1

...

Input: 010001 110101 011011
What is the output? Directly answer the question in the format 'Output:'.
```

- CoT prompt for DNF:

```
Replace some bits located at specific predefined positions in the binary string
↪  with 1 to form a new string. Then, based on some patterns in the new string
↪  to predict the output.

Input: 000101 011111 011010
New string: 100111 111111 011111
Output: 1

Input: 001101 100111 000101
New string: 101111 110111 000101
Output: 0

Input: 100001 011001 001010
New string: 100011 111011 001111
Output: 0

Input: 001100 010100 101011
New string: 101111 110110 101111
Output: 0

Input: 010101 111011 100101
New string: 110111 111011 100101
Output: 0
```

```
...
```

```
Input: 000100 011110 100000
What is the output? Directly answer the question in the format 'New string:,
↪  Output:'.
```

