# OpenReview forum: "An In-Context Learning Theoretic Analysis of Chain-of-Thought"
_ICML.cc/2024/Workshop/ICL — ICML 2024 Workshop ICL Poster_

### Official Review · Reviewer_oggB · 2024-06-07
**This paper develops a theoretical framework for understanding in-context learning (ICL) with Chain-of-Thought (CoT) prompts.**

**Rating:** 2
**Fit:** 3
**Confidence:** 2

**Workshop Review:**

This paper has two main contributions:

1. The paper introduces a Transformer construction capable of implementing different steps of ICL algorithms using CoT prompts. Each layer of the Transformer implements one step of gradient descent (GD) for a single intermediate step. Consequently, it requires $O(kt)$ layers to implement $t$-step GD for $k$-step CoT. The paper assumes that the nonlinearity of the data can be represented through kernel methods, allowing self-attention mechanisms to perform the GD algorithm.

2. The authors provide a theoretical analysis indicating that the performance of CoT is determined by the "worst" step, a concept that has been both empirically studied and theoretically analyzed in previous work. They also present upper and lower bounds for the "worst-case" error.

**Reason For Not Giving Higher Score:**

1. Despite focusing on general non-linear functions, the paper lacks discussion on how the proposed construction addresses non-linearity.
2. The proposed construction does not consider positional embeddings, necessitating a much higher model dimension, scaling as $O(kd)$ where $k$ is the number of intermediate CoT steps.
3. The paper lacks real-world experiments to validate the theoretical contributions.

**Reason For Not Giving Lower Score:**

1. The paper is well-organized and clearly presented.
2. The authors establish a framework to study ICL with CoT prompts and provide theoretical insights based on this framework.

---

### Official Review · Reviewer_dMzL · 2024-06-11
**Theoretical and empirical analyses of a particular instantiation of chain of thought reasoning**

**Rating:** 3
**Fit:** 3
**Confidence:** 2

**Workshop Review:**

This paper investigates a particular formalization of chain of thought reasoning, as a composition of nonlinear functions, where each function corresponds to one step of the chain of thought. And each step is optimized via gradient descent. The formalization makes sense as a potential algorithm, or at least a starting point for investigation. Especially since this algorithm actually is expressable by transformers, as shown by the authors.

The paper then follows up with theoretical analyses and empirical results, showing
* upper and lower bounds on the error
* implications for how to choose the intermediate steps (e.g. the hardest step is the bottleneck)
The empirical analyses are performed on interesting compositional tasks, e.g. boolean tasks and arithmetic tasks.

The work is well placed within related work, e.g. with connections to work on the relationship between in-context learning and gradient descent, or the relationship between transformers and kernel functions.

Perhaps the abstract and intro could make clearer that the algorithm is one particular formalization of chain-of-thought reasoning, not intended to be general. Renaming the algorithm from "ICL Algorithm" would also help with this.

**Reason For Not Giving Higher Score:**

n/a

**Reason For Not Giving Lower Score:**

This work is definitely worth sharing and discussing among those interested in in-context learning.

---

### Meta-Review · Area_Chair_DBea · 2024-06-14

**Recommendation:** 2

**Metareview:**

The paper presents a new theoretical framework to better understand in-context learning (ICL) using Chain-of-Thought (CoT) prompts. Following prior work, they provide weight constructions for transformers enabling them to learn via COT in a constrained setup.

Both reviewers agree on the quality and the fit of the paper to the ICL workshop. Reviewer dMzl points out that it's well placed within the related work, however also notes that the authors should be careful not to overclaim their contributions. Reviewer oggB similarly notes the good presentation of the paper and the establishment of a COT ICL framework, but points out that the work does not consider non-linear transformers nor positional embeddings.

Overall the paper is in good shape, provides solid contributions and hence I agree with the reviewers and accept the paper.

---

### Decision · Program_Chairs · 2024-06-17

Accept (Poster)